# Advanced Stiffness Sensing through the Pincer Grasping of Soft Pneumatic Grippers

**DOI:** 10.3390/s23136094

**Published:** 2023-07-02

**Authors:** Chaiwuth Sithiwichankit, Ratchatin Chanchareon

**Affiliations:** Department of Mechanical Engineering, Faculty of Engineering, Chulalongkorn University, Bangkok 10330, Thailand; chaiwuth.cs@gmail.com

**Keywords:** soft gripper, pneumatic gripper, pincer grasping, sensible grasping, stiffness sensing, object classification

## Abstract

In this study, a comprehensive approach for sensing object stiffness through the pincer grasping of soft pneumatic grippers (SPGs) is presented. This study was inspired by the haptic sensing of human hands that allows us to perceive object properties through grasping. Many researchers have tried to imitate this capability in robotic grippers. The association between gripper performance and object reaction must be determined for this purpose. However, soft pneumatic actuators (SPA), the main components of SPGs, are extremely compliant. SPA compliance makes the determination of the association challenging. Methodologically, the connection between the behaviors of grasped objects and those of SPAs was clarified. A new concept of SPA modeling was then introduced. A method for stiffness sensing through SPG pincer grasping was developed based on this connection, and demonstrated on four samples. This method was validated through compression testing on the same samples. The results indicate that the proposed method yielded similar stiffness trends with slight deviations in compression testing. A main limitation in this study was the occlusion effect, which leads to dramatic deviations when grasped objects greatly deform. This is the first study to enable stiffness sensing and SPG grasping to be carried out in the same attempt. This study makes a major contribution to research on soft robotics by progressing the role of sensing for SPG grasping and object classification by offering an efficient method for acquiring another effective class of classification input. Ultimately, the proposed framework shows promise for future applications in inspecting and classifying visually indistinguishable objects.

## 1. Introduction

One of the most important components of robotic systems is the gripper, performing as system hand to conduct grasping [1,2,3,4]. With grasping, the systems can have physical interaction with their environment and mechanically manipulate objects [5,6,7]. There are a variety of gripper types [8]. Different gripper types offer dissimilar grasping performances. Based on the principles of grasping, there are four generic types, including impactive, ingressive, astrictive, and contigutive [9]. The same source shows that impactive and astrictive grippers are the majority of grippers.

The human hand is always an extraordinary inspiration for impactive grippers in various aspects [10,11,12,13]. One of its motivating features is haptic sensing, allowing us to observe object properties and perform grasping adaptively [14]. Due to the rise of the complexity of robotic tasks, this feature has received considerable attention [15,16]. The sources moreover reveal that impactive grippers, the working principle of which is similar to the human hand, have been expected to make robotic systems capable of perceiving the mechanical properties of objects without any auxiliary probe. The systems with this capability are empirically more effective, in the aspects of working cost, space, and time, for object classification and identification [17,18,19,20,21], which are essential in an extensive scale of applications. This capability further facilitates adaptive grasping based on object properties [22,23].

Evidence shows that many researchers have studied the sensing of object properties through impactive grasping. However, available techniques for this function are for conventional impactive grippers with rigid links, broadly known as mechanical grippers. A variety of sensors were integrated into mechanical grippers to determine their internal states, external inputs, and loads [24,25,26,27]. For variables that cannot be directly perceived, kinematic and kinetic models of the grippers were developed. The association of variables on the grippers to those equivalents on grasped objects was solved, so object properties can be observed through gripper variables. Integrated sensors were mostly visual, force, and tactile types [28,29]. The following works are instances of the related studies. Kim [30] performed weight estimation using a force-sensor-integrated mechanical gripper. Romano et al. [31] implemented human-inspired adaptive grasping using a mechanical gripper with tactile sensors. Spiers et al. [32,33] studied feature extraction, shape recognition, and object classification using various mechanical grippers.

During the recent decades, a new gripper type, widely called a soft gripper, has increasingly obtained considerable interest [34,35,36]. Some soft grippers are of the impactive type. Their bodies are structurally compliant. The major outperforming feature of the grippers is mechanical intelligence, offering the possibility of grasping dissimilar objects using simple control architectures [37,38,39]. The soft pneumatic gripper (SPG), the working principle of which relies on anisotropic inflation according to pressurized air, is a popular type of soft gripper [40,41]. An SPG consists of two or more soft pneumatic actuators (SPAs). The main remarkable attribute of SPAs is that their intrinsically flexible structures can provide admirable compliance [42,43,44]. Nevertheless, the fantastic compliance of SPAs comes with a considerable concern regarding their nonlinear infinite degree-of-freedom (DOF) structures [45,46,47]. Accordingly, the modelling and sensor installation on SPAs are fundamentally challenging [48,49,50,51]. These challenges make the sensing of object properties through SPG grasping a complex problem. Specifically, the internal states and external loads of SPAs are difficult to perceive, and the connection of the variables on SPAs to those equivalents on grasped objects is mysterious.

Only a few relevant works were found. Homberg et al. [52] examined the data-driven haptic identification of objects using an SPG. Chen et al. [53] demonstrated size recognition and adaptive grasping using SPGs. Sankar et al. [54] investigated texture discrimination with an SPG. These works focus on the sensing of the sizes, geometries, and surface properties of grasped objects. Such attributes can be described with morphological variables, while mechanical properties require the consideration of both morphology and force. Supplementary studies are then necessary to make SPG grasping viable for the sensing of mechanical properties. At this early stage, a study relating to the sensing of any mechanical property through SPG grasping is a significant advancement.

A comprehensive framework for stiffness sensing through SPG pincer grasping is presented in this study. The presented framework includes the association of the deformation and force on the grasped object and their equivalents on SPAs, a new concept of SPA modeling, and a method for sensing object stiffness through SPG pincer grasping. The importance and originality of this study are that it explicates the combination of stiffness sensing and SPG grasping in a single action. The findings make an important contribution to the field of soft robotics, specifically extending the role of sensing for SPG grasping and object classification for complementing a new domain of classification data with an efficient framework of data acquisition. Future applications of the proposed framework could involve object inspection and classification for objects that cannot be discriminated based solely on their visual appearance.

## 2. Materials and Methods

### 2.1. Fundamentals of Stiffness Measurement

When an object is subjected to external force in the direction normal to its surface, it deforms in the same direction according to the force (Figure 1). Stiffness is a mechanical property of an object, indicating its ability to withstand deformation under external force. Let $\delta_{O}$ and $F_{O}$, respectively, represent the magnitude of the deformation and force on an object. Stiffness, denoted by $k_{O}$, governs the correlation between $\delta_{O}$ and $F_{O}$ as follows:(1)$${\partial F}_{O}=k_{O}\left(\delta_{O}\right)\cdot {\partial \delta }_{O}.$$Basically, $k_{O}$ is a function of $\delta_{O}$, and tends to be constant at low $\delta_{O}$. The conventional approaches for investigating $k_{O}$ are tensile [55] and compression [56] testing. An object is tensioned or compressed using a specific machine or probe, then a series of $\delta_{O}$ and $F_{O}$ during the test is collected. From the collected series of $\delta_{O}$ and $F_{O}$, $k_{O}$ can be determined following Equation (1). Likewise, objects are compressed by grippers throughout grasping. The mechanical interaction between the grippers and objects also causes $\delta_{O}$ and $F_{O}$. Accordingly, $\delta_{O}$ and $F_{O}$ throughout grasping can be utilized to estimate $k_{O}$. Thus, $k_{O}$ is alternatively realizable through grasping.

### 2.2. SPG Pincer Grasping

Impactive grasping can be basically categorized, based on morphology, into two main types, including pincer [57] and envelope [58] grasping. Objects are squeezed by the fingertips of grippers in pincer grasping. The contact between the grippers and the grasped objects certainly occurs around the fingertips, although the contact locations can be slightly deviated. In contrast, gripper fingers perform as cages confining objects in envelope grasping. The contact locations are unpredictable, and tactile sensors are required to determine where the contact takes place. Once the contact locations are determined, the analysis of the correlation between variables on the gripper fingers and their equivalents on the grasped objects can proceed. With these kind of correlations, $\delta_{O}$ and $F_{O}$ are solvable through the variables on the grippers, and then $k_{O}$ can be recognized.

This work focuses only on SPG pincer grasping (Figure 1). In SPG grasping, two or more SPAs perform as gripper fingers. SPAs make contact with the grasped objects at their tips. The interaction at the SPA tips depends on several factors, such as SPA performance, manipulator motion, object weight, and geometrical compatibility. To accomplish the sensing of $k_{O}$, grasping conditions must be properly set up to make other effects besides the interaction due to SPA performance negligible. In this work, manipulators were expected to be static, and object weight was presumed to be insignificant or compensated with supports. For geometrical compatibility, this factor is supposed to have no effect on the interaction according to SPA compliance. Our observation and analysis on the pincer grasping of two-finger impactive SPGs reveals that $F_{O}$ equals the magnitude of the withstanding force on the SPAs, denoted by $F_{S}$ (Figure 1), as follows:(2)$$F_{S}\left(t\right)=F_{O}\left(t\right).$$The deformation of the SPAs, denoted by $\delta_{S}$ (Figure 1), is equivalent to the combination of $\delta_{O}$ and the space between them, denoted by G (Figure 1), as follows:(3)$$\delta_{S}\left(t\right)=\delta_{O}\left(t\right)+G.$$

### 2.3. SPA Modeling

The infinite-DOF structures of SPAs not only complicate the connection of $\delta_{S}$ and $F_{S}$ to $\delta_{O}$ and $F_{O}$, but also make SPA modeling difficult. The modeling of SPAs is still an area of mainstream research on soft robotics. The most extensive technique for managing SPA infinite-DOF structures has been utilizing the combined effect of individual local joints to depict SPA configurations. In published SPA models, SPA curvature was assumed to be consistent along SPA length. SPA configuration was then represented by the SPA angle of curvature [50,59]. Nonetheless, SPA curvature is practically variable along the length in SPG pincer grasping. We suggest that SPA configuration should be portrayed by $\delta_{S}$. According to the equation of motion, an SPA model can be initiated as follows:(4)$${\ddot{\delta }}_{S}\left(t\right)=f(\delta_{S}\left(t\right),{\dot{\delta }}_{S}\left(t\right),F_{S}\left(t\right),P_{S}\left(t\right)),$$
where $P_{S}$ denotes SPA input pressure. The variables ${\dot{\delta }}_{S}$ and ${\ddot{\delta }}_{S}$ are time derivatives of $\delta_{S}$, while f represents a nonlinear function due to SPA characteristics. Equation (4) indicates that the change in ${\ddot{\delta }}_{S}$ is governed by $\delta_{S}$, ${\dot{\delta }}_{S}$, $F_{S}$, and $P_{S}$. However, SPA conditions are mostly static throughout grasping. Both ${\dot{\delta }}_{S}$ and ${\ddot{\delta }}_{S}$ are then typically negligible. Equation (4) can be revised as follows:(5)$$0=f\left(\delta_{S}\left(t\right),F_{S}\left(t\right),P_{S}\left(t\right)\right).$$This equation shows that $\delta_{S}$ can be repeated under the identical steady-state conditions of $F_{S}$ and $P_{S}$.

According to published works on SPA modeling, solving Equation (5) is demanding. There is no straightforward analytical method for it, and available simulation solvers are not suitable because of the large deformation and nonlinear characteristics of SPAs. An empirical investigation on an SPA was consequently conducted to obtain its performance curves, serving as an SPA model for comprehensive demonstration. Note that the SPA is an industrial bending type, and its operating $P_{S}$ condition ranges from −100 to 100 kPa. The experimental setup used for the investigation includes a motorized linear stage, load cell, and air supply, as demonstrated in Figure 2. The SPA was hung vertically. The load cell was equipped to the linear stage, and then horizontally moved to constrain $\delta_{S}$. The air supply was later employed to generate $P_{S}$. After the SPA became steady, the signal from the load cell was interpreted to measure $F_{S}$. We subsequently iterated these steps for different combinations of $\delta_{S}$ and $P_{S}$ until $F_{S}$ at all designated $\delta_{S}$ and $P_{S}$ conditions was collected and recorded. Note that the designated conditions of $\delta_{S}$ ranged from 0 to 8 mm with 200 µm intervals, and those of $P_{S}$ ranged from 10 to 100 kPa with 10 kPa intervals. Linear fitting using QR factorization [60] was then applied to the resulting relationship between $\delta_{S}$ and $F_{S}$ at each $P_{S}$ condition. Finally, the fitting results were plotted as $F_{S}$–$\delta_{S}$ straight lines.

### 2.4. Stiffness Sensing through SPG Pincer Grasping

The proposed method of sensing $k_{O}$ is basically based on object compression due to SPA performance throughout SPG pincer grasping. Equations (2) and (3) illustrate that $\delta_{O}$ and $F_{O}$ can be algebraically transformed into $\delta_{S}$ and $F_{S}$. Equation (1) is then revisable in accordance with the association of $\delta_{S}$ and $F_{S}$ to $\delta_{O}$ and $F_{O}$ as follows:(6)$${\partial F}_{S}=k_{O}\left(\delta_{S}-G\right)\cdot \partial \delta_{S}.$$From this perspective, $k_{O}$ can be alternatively determined from some simultaneous sets of $\delta_{S}$ and $F_{S}$, producible by varying $P_{S}$. The sensing of $\delta_{S}$ and $F_{S}$ is still demanding. Most of the established works on SPA sensing in the aspect of SPA configuration only consider the SPA angle of curvature, which cannot be uniquely transformed into $\delta_{S}$ in SPG pincer grasping. Despite sensing accuracy, we found that visual inspection is the only available approach for the direct observation of $\delta_{S}$. Furthermore, the sensing of $F_{S}$ can proceed only when using cutting-edge tactile sensing [26,61]. Currently, the most straightforward method for determining both $\delta_{S}$ and $F_{S}$ is through exploiting computer vision to visually inspect $\delta_{S}$. Once $\delta_{S}$ is successfully perceived, and $P_{S}$ is identified using some extensive pressure sensors, $F_{S}$ can be approximated using SPA kinetic models. The method of solving of Equation (5) mentioned in the previous subsection is presented for this purpose.

Stiffness sensing through SPG pincer grasping on four testing samples, subsequently called thin, slim, medium, and thick sample, was carried out. The testing samples were elastic elastomer hollow tubes (Figure 3), 3D-printed from the same material using a fused deposition modeling 3D printer with an extrusion nozzle of 0.4 mm. Each of them weighs around 2 g and are thus negligible. Their outer diameter was designed to be 20 mm. The main difference among them is their wall thickness. The wall thickness of the thin sample was 0.8 mm, that of the slim sample was 1.2 mm, that of the medium sample was 1.6 mm, and that of the thick sample was 2 mm. The SPG utilized in this experiment consisted of two SPAs, identical to the one investigated in the previous subsection, with a space of 20 mm between them. One of the samples was instantly grasped in a vertical gesture (Figure 4) under 11 different conditions of $P_{S}$ ranging from 0 to 100 kPa with 10 kPa intervals. At each $P_{S}$ condition, a stereo photograph of grasping was captured when the steady state was reached using a binocular digital camera, the resolution of which was FHD. The loaded $P_{S}$ was then released to make the SPG and sample return to their neutral shapes, before the SPG was executed with another of the $P_{S}$ conditions. After all of the stereo photographs were collected, each of them was individually processed to estimate G and $\delta_{S}$ using image processing [62,63] and stereo computer vision [64,65]. From the estimated $\delta_{S}$ and realized $P_{S}$, the corresponding $F_{S}$ was identified. The correlation between $\delta_{S}$ and $F_{S}$ at every designated $P_{S}$ condition was obtained accordingly. The next steps were eliminating the data points at which $\delta_{S}$ was below G or beyond 10 mm, and then adding the *x*-axis intersection point at (G,0). Linear interpolation was utilized to make the discrete points into a continuous line. Hence, $k_{O}$ could be determined as the slope of the $F_{S}$–$\delta_{S}$ curve following Equation (6). The resulting $F_{S}$–$\delta_{S}$ curve was alternatively converted into an $F_{O}$–$\delta_{O}$ curve using Equations (2) and (3), and $k_{O}$ was computed following Equation (1). The same process was repeated with the other samples.

In particular, the workflow of image processing and stereo computer vision began with the image undistortion [66] on a pair of the stereo photographs using pre-calibrated intrinsic camera parameters. The SPA tips were manually extracted from the undistorted images, and then the point triangulation [67] on the extracted pixels was completed. From the point triangulation, the 3D positions, with respect to the camera, of the SPA tips were determined. The vector from one SPA tip to the other was formulated in 3D based on their determined positions, and the distance between the SPA tips was calculated as the magnitude of this vector. These procedures were repeated for every pair of photographs. At this point, the distance between the SPA tips at every predefined $P_{S}$ condition was estimated. The distance at the rest state was set as a reference, and other estimated distances were compared to it. The resulting margin was considered twice $\delta_{S}$. For the value of G, we manually extracted the edges of the sample from one of the photograph pairs. Next, the extracted pixels were processed through the same procedures to determine the outer diameter of the sample. The approximated diameter was subtracted from the distance between the SPA tips at the rest state, and the subtraction result was considered twice G. The process toward the distance was tested on the exterior size of the samples, and it caused errors of up to 200 µm.

### 2.5. Validation

To validate the proposed method of stiffness sensing, a compression-testing experiment was conducted on the testing samples. Each sample was compressed individually using a load cell driven with a motorized linear stage (Figure 5). Specifically, the capacity of the load cell was 1 kgf, and its accuracy was 0.2% of the capacity. We adjusted $\delta_{O}$ by moving the load cell, and compressed the sample until it reached a steady state. Later, $F_{O}$ was measured with the load cell. These steps were repeated for $F_{O}$ at the $\delta_{O}$ conditions ranging from 0 to 6 mm with 100 µm intervals. From the acquired pairs of $\delta_{O}$ and $F_{O}$, a curve of $F_{O}$ over $\delta_{O}$ was plotted. This curve was then transformed into $k_{O}$ over $\delta_{O}$ using Equation (1). Afterwards, the resulting $k_{O}$ over $\delta_{O}$ was compared to its counterpart obtained via the proposed method. Specifically, the comparison was focused on the deviation of $k_{O}$, denoted by ${∆k}_{O}$, along $\delta_{O}$. The definition of ${∆k}_{O}$ was determined as follows:(7)$${∆k}_{O}\left(\delta_{O}\right)=\left|k_{O,S}\left(\delta_{O}\right)-k_{O,C}\left(\delta_{O}\right)\right|,$$
where $k_{O,S}$ and $k_{O,C}$, respectively, represent $k_{O}$ gathered from SPG pincer grasping and compression testing. These procedures were iterated for all other samples. In addition, the results of the proposed method were verified, and they were viable for object classification by utilizing them in manual template-matching classifications. The template used in this process is $k_{O}$ over $\delta_{O}$ obtained through compression testing.

## 3. Results and Discussion

### 3.1. SPA Modeling

The empirically-investigated model of an SPA was first obtained. It contains 10 performance curves (Figure 6), describing SPA behaviors at 10 $P_{S}$ conditions ranging from 10 to 100 kPa with 10 kPa intervals. Each curve addresses the inverse relationship between $\delta_{S}$ and $F_{S}$ at a specified $P_{S}$ condition. Furthermore, both $\delta_{S}$ and $F_{S}$ were raised when $P_{S}$ increased. The resolution of $\delta_{S}$ intervals in the investigation had only a minor effect on the resulting curves. The decrease rate of $F_{S}$ with respect to $\delta_{S}$ varied from around 261 to 415 N/m. We tried the $\delta_{S}$ intervals of 100 and 400 µm for the same process, and there was a tiny change in the results. The processing effort and time required for the investigation can be considerably reduced by using a rougher $\delta_{S}$ interval. Another observation of this examination was that SPA curvature was practically inconsistent along SPA length (Figure 7) when $F_{S}$ was plentiful. The need for a new concept of SPA modeling is hence remarked upon by this investigation.

### 3.2. Stiffness Sensing through SPG Pincer Grasping

The experiment of stiffness sensing through SPG pincer grasping was completed. Eleven stereo photographs of SPG pincer grasping (Figure 7), captured at $P_{S}$ conditions ranging from 0 to 100 kPa with 10 kPa intervals, were acquired for each tested sample. The expressed workflow of image processing and stereo computer vision utilized these photographs and provided the distance between SPA tips at every $P_{S}$ condition. A set of $\delta_{S}$ with respect to $P_{S}$ was determined from the provided distances. The corresponding $F_{S}$ was realized from the obtained pairs of $\delta_{S}$ and $P_{S}$ using the SPA model acquired in the previous subsection. The resulting correlation between $\delta_{S}$ and $F_{S}$ was made continuous using piecewise linear interpolation (Figure 6). Through the same workflow of image processing and stereo computer vision, the value of G was found to be 1.44 mm for the thin sample, 1.38 mm for the slim sample, 1.27 mm for the medium sample, and 1.4 mm for the thick sample. The equivalent values of $\delta_{O}$ and $F_{O}$ were calculated using Equations (2) and (3). Four $F_{O}$–$\delta_{O}$ curves (Figure 8) were plotted following the calculated collection of $\delta_{O}$ and $F_{O}$. The stiffness $k_{O}$ of all samples was plotted (Figure 8), and therefore successfully perceived through SPG pincer grasping. We found that $k_{O}$ was flat between 56 and 147 N/m for the thin sample. For the slim sample, $k_{O}$ was plane between 188 and 263 N/m when $\delta_{O}$ was below 4 mm and continuously increased to 973 N/m when $\delta_{O}$ became greater. Furthermore, $k_{O}$ increased from 308 to 562 N/m when $\delta_{O}$ was smaller than 1.4 mm, and was fairly horizontal between 442 and 626 N/m when $\delta_{O}$ was greater for the medium sample. Lastly, $k_{O}$ was found to be consistent at around 1.1 N/mm when $\delta_{O}$ was under 500 µm, and then oscillated between 1.1 and 2.1 N/mm when $\delta_{O}$ was higher than 500 µm for the thick sample.

### 3.3. Validation

Compression testing on the testing samples was successful, and four $F_{O}$–$\delta_{O}$ curves were collected (Figure 8). The value of $k_{O}$ was subsequently computed and plotted (Figure 8). The resulting $k_{O}$ from the both methods had similar trends with small deviations for all samples. The deviation ${∆k}_{O}$ with respect to $\delta_{O}$ was computed (Figure 8). The comparison results show that ${∆k}_{O}$ was oscillating over $\delta_{O}$ around 40 N/m for the thin sample when $\delta_{O}$ was under 5 mm, and continuously rose to 200 N/m when $\delta_{O}$ was higher for the thin sample. Meanwhile, ${∆k}_{O}$ was oscillating around 100 N/m when $\delta_{O}$ was below 4 mm, and suddenly shifted to vibrate around 25 to 30 N/m when $\delta_{O}$ went over 4 mm for the slim sample. For the medium sample, ${∆k}_{O}$ was roughly flat around 150 N/m when $\delta_{O}$ was between 1.38 and 2.9 mm, while fluctuating around 300 N/m when $\delta_{O}$ was out of this range. Moreover, ${∆k}_{O}$ was oscillating around 400 N/m for the thick sample. The viability of $k_{O}$ acquired through SPG pincer grasping for object classification was also tested by employing the results of $k_{O}$ for manual template-matching classification on the samples. All of the samples were promptly classified because the classifying data obviously fit the template. The consideration was mainly based on the maximum value and moving average of $k_{O}.$ The classification results are demonstrated in Figure 9. Although the classification was not programmatically processed, it illustrates that $k_{O}$ obtained with the proposed method can be employed for object classification.

In the cases of the thin and slim samples, ${∆k}_{O}$ dramatically increased when $\delta_{O}$ was greater than 4 mm. This phenomenon could be attributed to the occlusion effect [68], which occurred during manual feature extraction in the process of stereo computer vision. The phenomenon obstructed the visibility of the SPA tips when $\delta_{O}$ yielded 4 mm. Once the SPA tips were unreliably extracted, the estimated values of $\delta_{S}$ could deviate from the real values, and the resulting $k_{O}$ from the proposed method could have considerable errors. The necessity of further research on direct $\delta_{S}$ sensing for more accurate results of $k_{O}$ is hence remarked. With more accurate $\delta_{S}$, the proposed method could offer more rigorous results of $k_{O}$, and be credibly applied to some higher level tasks, such as object identification based on $k_{O}$.

## 4. Conclusions

The sensing of stiffness through the pincer grasping of SPGs is presented in this study. The main objective of this study was to clarify the sensing of object stiffness via SPG grasping. The challenge of the study was that the deformation and force on the grasped objects must be associated with their equivalents on SPAs, but SPA structure basically has infinite DOF with nonlinear characteristics. According to SPA compliance, the modeling and sensor installation on SPAs are also sophisticated and novel. We first clarified how object stiffness is conventionally measured, and then presented the connection between stiffness measurement and SPG grasping. The analysis on SPG grasping was completed. Next, the association of the deformation and force on grasped objects to those on SPAs during SPG pincer grasping was formulated. Because SPA curvature is practically inconsistent along SPA length in SPG pincer grasping, available methods of SPA modeling are unfunctional in this case. A new concept of SPA modeling was consequently introduced. Afterwards, a method for stiffness sensing through SPG pincer grasping was proposed and demonstrated on four testing samples. An experiment of compression testing was later implemented to validate the results of the proposed method. The results show that the stiffness of all of the samples perceived with this proposed method slightly deviated from that obtained from compression testing; however, it was still functional for classifying the samples. The contribution and novelty of this study are combining stiffness sensing and SPG grasping into a single action. This study contributes to the fields of soft robotics by leading to the role extension of SPG grasping for sensing applications and object classification by offering an efficient framework for acquiring a new class of classification inputs. The proposed framework has potential for future applications in object inspection and classification, especially for visually indistinguishable objects. A limitation of this study is that the resulting deviation dramatically increased when the deformation of a sample yielded a large value. This phenomenon could be explained by the occlusion effect during the process of visual inspection for SPA deformation. Further studies regarding the direct sensing of SPA deformation should be carried out to overcome the occlusion effect and substantially improve the resulting accuracy of the proposed method. The accuracy improvement is expected to make the outputs of the proposed method viable for higher level tasks, such as identifying grasped objects.

## Figures and Tables

**Figure 1 sensors-23-06094-f001:**
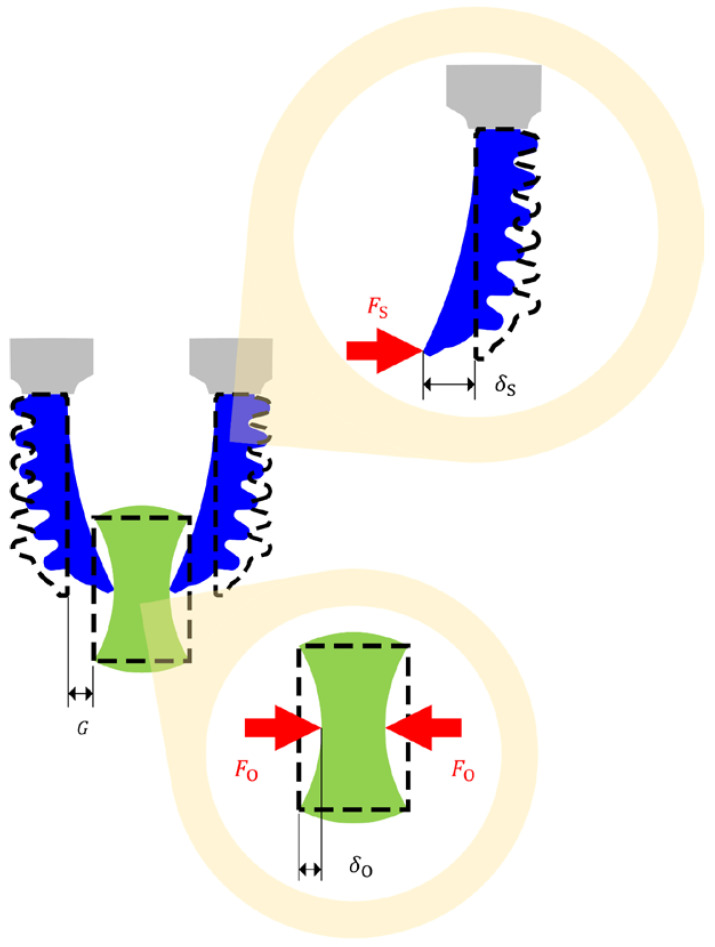
SPG grasping an object with deformation and force annotations on the SPA and object.

**Figure 2 sensors-23-06094-f002:**
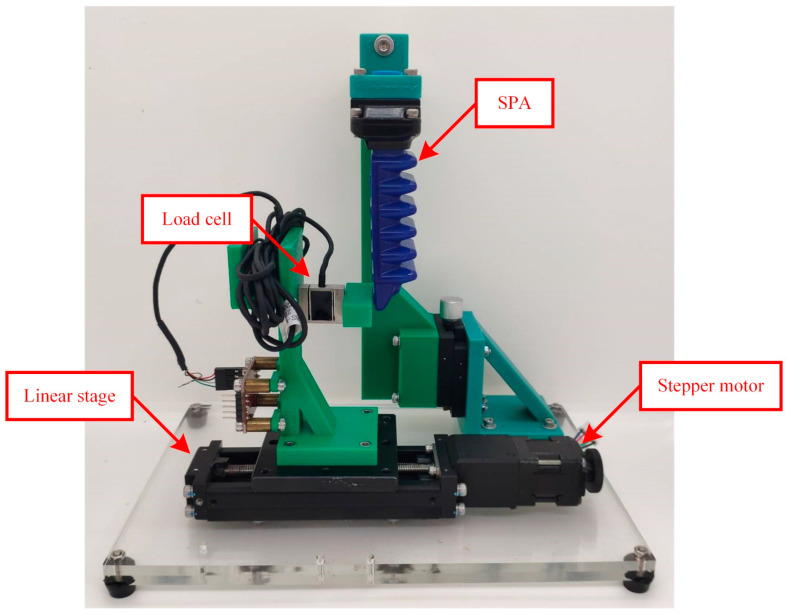
An experimental setup for acquiring SPA performance curves.

**Figure 3 sensors-23-06094-f003:**
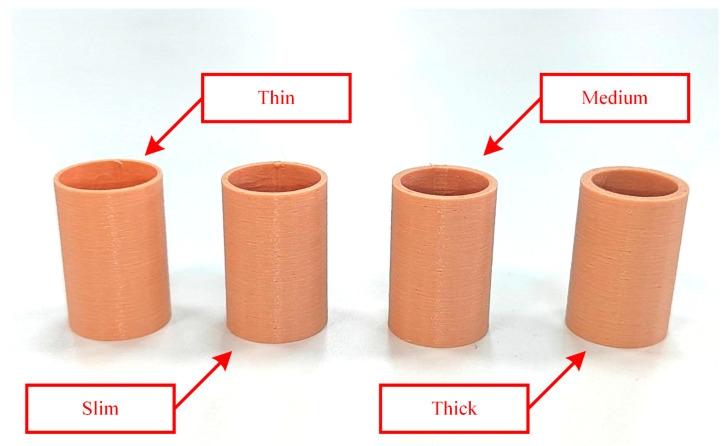
Elastic elastomer 3D-printed hollow tubes used as testing samples for stiffness recognition.

**Figure 4 sensors-23-06094-f004:**
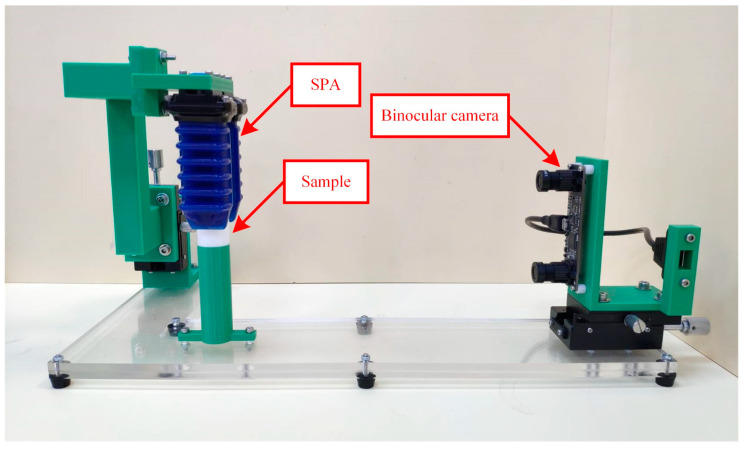
Experimental setup for stiffness sensing through SPG pincer grasping.

**Figure 5 sensors-23-06094-f005:**
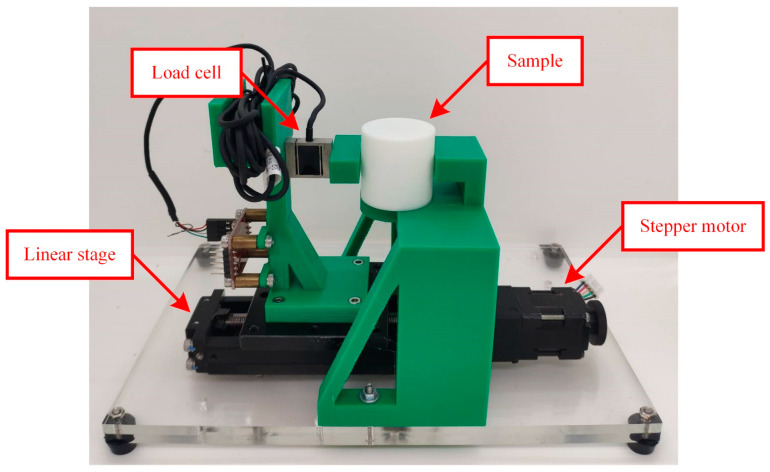
Experimental setup for stiffness measurement through compression testing.

**Figure 6 sensors-23-06094-f006:**
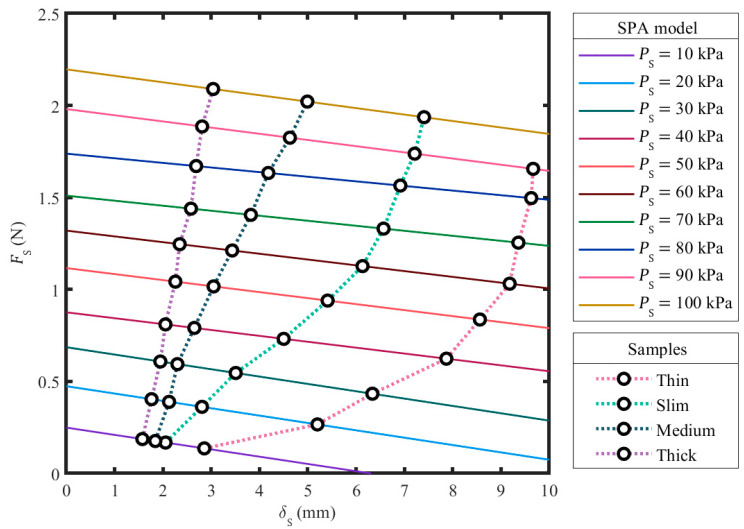
An empirical kinetic model of an SPA and the primary results of stiffness sensing through SPG pincer grasping on four samples.

**Figure 7 sensors-23-06094-f007:**
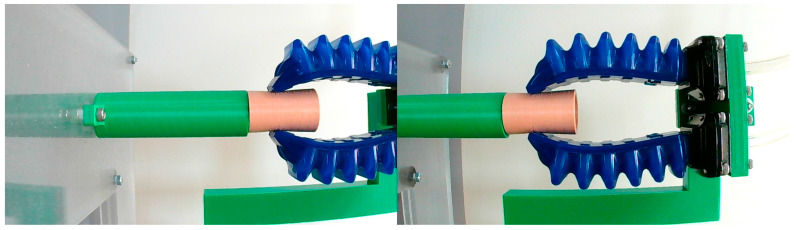
A stereo photograph of SPG grasping captured with a binocular camera.

**Figure 8 sensors-23-06094-f008:**
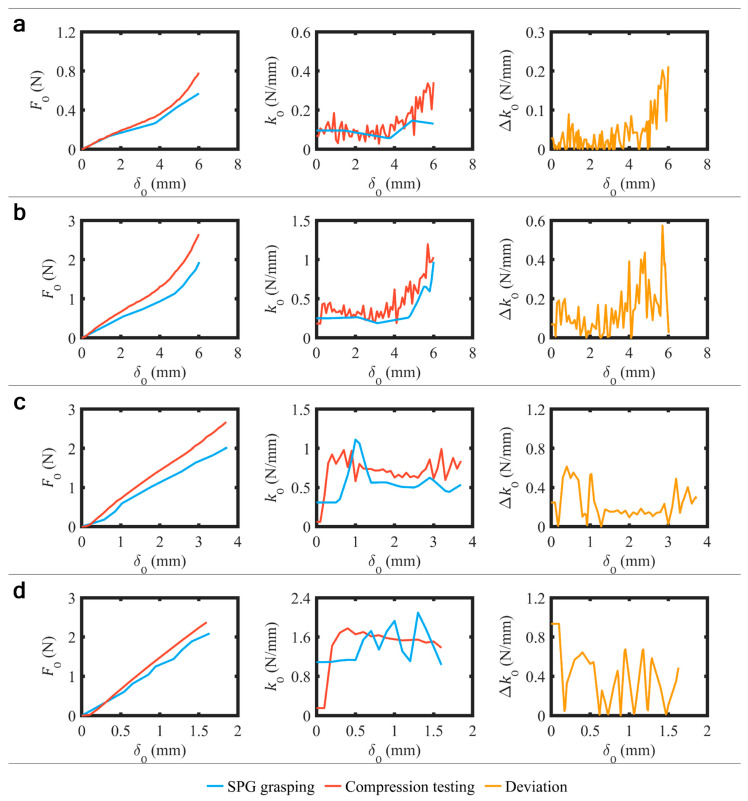
Resulting force, stiffness, and stiffness deviations over object deformation from stiffness sensing through SPG pincer grasping and compression testing methods on four samples: (**a**) thin sample; (**b**) slim sample; (**c**) medium sample; (**d**) thick sample.

**Figure 9 sensors-23-06094-f009:**
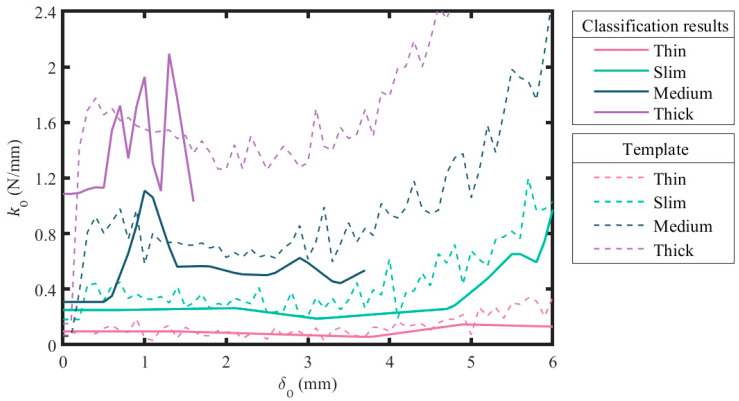
Manual template-matching classification on four samples based on their stiffness collected through SPG pincer grasping.

## Data Availability

Not applicable.

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
