# Peer review of "Advanced Stiffness Sensing through the Pincer Grasping of Soft Pneumatic Grippers"

_sensors, 2023, doi:10.3390/s23136094_

Round 1

Reviewer 1 Report

Paper sensors-2449301 presents a comprehensive approach for sensing object stiffness through pincer grasping of soft pneumatic grippers. The paper contents are adequate to the scope of Sensors. There are, however, several factors that prevent the publication of this article in its present form.

1. The abstract section is poorly written and lacks contextualization. The content is also simply listed, and the text is not fluent enough.

2. It is suggested to add the innovation points of this paper in the introduction section.

3. Image 1 is too large, suggest reworking to fit journal specifications.

4. Introduction: ‘This capability furthermore facilitates further adaptive grasping based on model-based control.’ There are some logical problems with the sentence, which suddenly leads to "model-based research".

5. ‘Note that the designated conditions of ?S ranged from 0 to 8 mm with 200-µm intervals, and those of ?S ranged from 10 to 100 kPa with 10 kPa intervals.’ How is the test range determined? The article does not indicate the size of the SPA used.

6. In Figure 6, Sample#2 shows a large change in slope for Ps 70-100kPa. What is the reason for this problem?

7. Conclusions: please point out more specific scenarios and prospects for the application of the method in the future.

8. There is no obvious comparison seen in the article and it does not feel convincing enough to have a quantitative description.

Abstracts need to be reorganized to improve the readability of the text.

Reviewer 2 Report

The authors describe Advanced Stiffness Sensing Through Pincer Grasping of SPG. The paper presents the theoretical part describing the forces and deformations that arise during the use of the grip with the object. The theoretical part is similar to the article "Adaptive Pincer Grasping of Soft Pneumatic Grippers Based on Object Stiffness for Modellable and Controllable Grasping Quality". Selected drawings appear in both manuscripts. In the content of the manuscript, the authors present that 2 types of samples were used for the study. One sample, the wall thickness of sample No. 1 is 0.8 mm, and sample No. 2 is 1.2 mm, the diameter is therefore 20 mm. In the article, the authors do not describe the reasons why the walls are dimensioned (it can be concluded that it is about a change in stiffness, but the question arises why not 0.5mm wall thickness?), it should be justified so that the reader has no doubts. Measurement of deformations and changes in dimensions based on photos raises some doubts. The article does not say how exactly the distance was calculated in the photos, there is no scale, e.g. in mm, and how to avoid measurement errors (these are key parameters that affect the results and conclusions). The content of the article contains information about SPA modeling. The drawings and formulas describe the forces acting (actions and reactions without taking into account the force of gravity or the surface structure of the object, the force of friction) as well as deformations of the object and grippers. The weight was omitted because the samples had a few grams, but in real conditions it will be a much larger weight that should be taken into account because it will affect the gripper and generate additional resultant forces. The modeling should reflect the conditions of the actual SPA operation, either in graphical form or numerical simulation. The manuscript describes a solution that takes a different approach to detecting stiffness, but needs tweaking.

English is understandable.

Round 2

Reviewer 1 Report

The comments and questions raised by the reviewers have been addressed. This manuscript can be accepted in its current form.

Minor editing of English language required.

Reviewer 2 Report

The manuscript presents a theoretical part describing the forces and deformations that arise when gripping an object. The authors explain why the dimensions of the samples have a certain thickness (3D printout). The distance and deformation measurement method has been partially described. The details shown in Figure 1 are the same in two identical drawings (the one above the corner looks doubled). The authors increased the number of samples tested, which increased the attractiveness of the manuscript and allowed to extend the obtained results and formulate more extensive conclusions.

Minor editorial errors that do not affect the quality or comprehensibility of the manuscript content.